# Characterization of Long Non-Coding RNAs in the Bollworm, *Helicoverpa zea*, and Their Possible Role in Cry1Ac-Resistance

**DOI:** 10.3390/insects13010012

**Published:** 2021-12-22

**Authors:** Roger D. Lawrie, Robert D. Mitchell, Jean Marcel Deguenon, Loganathan Ponnusamy, Dominic Reisig, Alejandro Del Pozo-Valdivia, Ryan W. Kurtz, Richard Michael Roe

**Affiliations:** 1Department of Entomology and Plant Pathology, North Carolina State University, Campus Box 7647, 3230 Ligon Street, Raleigh, NC 27695, USA; rdlawrie@ncsu.edu (R.D.L.); Mitchell.Robert@epa.gov (R.D.M.III); jdeguen@ncsu.edu (J.M.D.); loganathan_ponnusamy@ncsu.edu (L.P.); 2Environmental and Molecular Toxicology Program, Department of Biology, College of Sciences, North Carolina State University, 2601 Stinson Drive, Raleigh, NC 27606, USA; 3Office of Pesticide Programs, Invertebrate and Vertebrate Branch 1, Registration Division, U.S. Environmental Protection Agency, 1200 Pennsylvania Avenue, Washington, DC 20460, USA; 4Vernon G. James Research & Extension Center, Department of Entomology and Plant Pathology, 207 Research Station Road, Plymouth, NC 27962, USA; ddreisig@ncsu.edu (D.R.); adelpozo@ucanr.edu (A.D.P.-V.); 5Cotton Incorporated, 6399 Weston Parkway, Cary, NC 27513, USA; RKurtz@cottoninc.com

**Keywords:** long non-coding RNAs, *Helicoverpa zea*, Bt-resistance, Cry1Ac resistance, RNA-seq, lncRNA, bollworms, gene regulation

## Abstract

**Simple Summary:**

Multiple caterpillar pest species have become resistant to transgenic crops. These crops were originally engineered to make a bacteria protein that would kill the caterpillar when the insect eats the plant. This project focused on comparing gene expression patterns in a bollworm caterpillar resistant strain compared to a susceptible bollworm strain. Expression differences were found in long non-coding RNAs, sequences that do not make proteins but can regulate making proteins. There were increased and decreased levels of different long non-coding RNAs in the resistant strain. Proximity relationships of these non-coding RNAs to protein coding-genes that have functions known to cause resistance were also found. Proximity is one way long non-coding RNA regulates the making of proteins and could be a mechanism of how these insects became resistant. The potential of using these discoveries in managing insect pest resistance levels in the field is discussed.

**Abstract:**

Multiple insect pest species have developed field resistance to Bt-transgenic crops. There has been a significant amount of research on protein-coding genes that contribute to resistance, such as the up-regulation of protease activity or altered receptors. However, our understanding of the role of non-protein-coding mechanisms in Bt-resistance is minimal, as is also the case for resistance to chemical pesticides. To address this problem relative to Bt, RNA-seq was used to examine statistically significant, differential gene expression between a Cry1Ac-resistant (~100-fold resistant) and Cry1Ac-susceptible strain of *Helicoverpa zea*, a prevalent caterpillar pest in the USA. Significant differential expression of putative long non-coding RNAs (lncRNAs) was found in the Cry1Ac-resistant strain (58 up- and 24 down-regulated gene transcripts with an additional 10 found only in resistant and four only in susceptible caterpillars). These lncRNAs were examined as potential pseudogenes and for their genomic proximity to coding genes, both of which can be indicative of regulatory relationships between a lncRNA and coding gene expression. A possible pseudogenic lncRNA was found with similarities to a cadherin. In addition, putative lncRNAs were found significantly proximal to a serine protease, ABC transporter, and CYP coding genes, potentially involved in the mechanism of Bt and/or chemical insecticide resistance. Characterization of non-coding genetic mechanisms in *Helicoverpa zea* will improve the understanding of the genomic evolution of insect resistance, improve the identification of specific regulators of coding genes in general (some of which could be important in resistance), and is the first step for potentially targeting these regulators for pest control and resistance management (using molecular approaches, such as RNAi and others).

## 1. Introduction

In integrated pest management (IPM) practices, an effective method of pest control for many years has been Bt (*Bacillus thuringiensis*)-transgenic crops. Insecticidal proteins (including Cry family proteins) isolated from this bacteria have been cloned into commercial crops (corn, soybeans, cotton, etc.) and have been successful in the control of insect pest species, such as lepidopterans [1]. Unfortunately, rapidly increasing levels of resistance have been detected worldwide in multiple pest species in recent years. These include but are not limited to populations of the fall armyworm, *Spodoptera frugiperda*, in Puerto Rico and North Carolina (USA); the maize stalk borer, *Busseola fusca*, in South Africa; the pink bollworm, *Pectinophora gossypiella*, in India and the USA; and the bollworm, *Helicoverpa zea*, in the USA [2,3,4,5,6,7,8,9,10]. *H. zea,* the model organism in this study, has been reported to have resistance to multiple Cry family proteins, including Cry1Ac, Cry1A.105, Cry1Ab, Cry2Ab2, and Cry1F [8,9,10,11]. *H. zea* is a prevalent polyphagous pest that annually causes significant economic damage to crops [8,9,10,11]. It has also been noted that resistance to Cry1A family proteins has plateaued in *H. zea,* but Cry2A family resistance is still being selected [11]. Recently in *H. zea*, resistance to Vip3Aa was detected in the USA, the magnitude of which varies in individual populations [12,13].

There are multiple recognized mechanisms of Bt-resistance in insects. These are alterations in serine proteases, Bt-receptors in the gut cavity (cadherin, alkaline phosphatase, and aminopeptidase), transporters (ABC transporters), tetraspanin, secretase, and trypsin proteins [14,15,16,17,18,19,20,21]. Our research team recently reported on the potential involvement of the insect immune system in Bt-resistance and changes in P450s [22]. Bt-receptors have been linked to resistance via mutations or altered binding of Bt-proteins to the mid-gut receptors [23,24]. It should also be noted that the ABCC2 gene has been shown to not act as a Bt-receptor and that alkaline phosphatase 2 is attributed to Cry1Ac but not Cry2Ab resistance [25,26]. Yang et al. (2020) reported Vip3Aa resistance inheritance was monogenic, autosomal, and recessive, but the exact gene annotation is unclear [12]. An understudied area of Bt-resistance is the involvement of non-protein-coding genetic elements.

Long non-coding RNAs (lncRNAs) are RNA strands over 200 nucleotides in length that are not translated into proteins but are structurally similar to mRNA [27]. Mechanistically, lncRNAs can act in a variety of roles in regulating coding genes. LncRNAs can affect *cis* or *trans* gene expression, transcription factor activation or repression through binding and localization, chromatin remodeling, imprinting, and enhancer regulation [27]. They are also involved in broader regulatory processes, such as post-transcriptional regulation, protein trafficking, and mRNA processing [27]. Chromatin remodeling activity occurs via lncRNA mediated histone methylation, which leads to a shift in chromatin structure, either increasing or decreasing access of transcriptional machinery to DNA strands [28]. In this role, the lncRNA acts as a protein scaffold with several binding domains allowing proteins involved in methylation and demethylation to interact with the target histone [28]. Transcriptional suppression can also occur when a lncRNA acts as a “decoy” for RNA polymerases by binding the polymerase to a lncRNA instead of the normal target DNA [29]. LncRNAs can also regulate translation by binding to mRNAs, increasing translation, decreasing translation, or promoting the bound mRNA’s degradation [30].

LncRNAs have been studied most in humans and animals used in medical research. Research is minimal by comparison in insects, even less for agricultural pests, and there have only been a few studies on their role in pesticide resistance (for example, one in *Plutella xylostella*) [31]. The lncRNA regulatory role in insect transcriptional processes and their relationship to insecticide resistance was recently studied [31]. In honeybees, *Apis mellifera,* high expression levels of lncRNAs were found in ovaries, likely because they play a role in developmental processes, such as social caste determination [32]. In fall armyworms, *Spodoptera frugiperda*, lncRNA expression was correlated to heterochromatin formation [33]. In *Drosophila*, lncRNAs appeared to coordinate sex determination [34].

There is recent evidence that lncRNAs are involved in insecticide resistance, increased fitness, and responsiveness to xenobiotic exposure. For example, lncRNAs were associated with chlorpyrifos insecticide resistance in the diamondback moth, *Plutella xylostella.* That study found that lncRNA overexpression likely regulated the increased expression of resistance-associated genes, such as those that code for detoxifying enzymes [31]. The brown rice plant hopper, *Nilaparvata lugens*, has become rapidly resistant to many different insecticides, with high fecundity and virulence compared to the susceptible strain. It was discovered that significant differences existed in the lncRNA profiles between the two strains, suggesting that lncRNAs had a role in resistance [35]. In the pink bollworm, a specific lncRNA was responsible for transcriptional regulation of *P. gossypiella* cadherin 1 (*pgCad1),* which encodes for a midgut receptor known to be involved in Bt-resistance. Using RNAi (RNA interference) that targeted the lncRNA, larval Bt-toxin susceptibility was altered [36].

The objective of this study was to examine the role of lncRNAs in Bt-resistance in the bollworm, *H. zea*. The insects used in this study were Cry1Ac-resistant and Cry1Ac-susceptible (referred to later as Bt-resistant or Bt-susceptible for brevity). Additionally, this study aims to increase the overall understanding of the functional role of lncRNAs in insects. A shotgun RNA-seq approach was used to compare the gene expression profiles of a Bt-resistant and a Bt-susceptible strain of bollworm. From this whole transcriptome dataset, putative lncRNA sequences were isolated for analysis. The potential for pseudogenes in this dataset was also explored, which can be a source from which lncRNAs are derived. Additionally, the relationship of putative lncRNAs in genomic proximity (defined as within 1 million bases) to coding genes was analyzed. This study is a first step in characterizing these formerly uncharacterized portions of the *H. zea* genome.

## 2. Methods

### 2.1. Sample Collection and Preparation

*Helicoverpa zea* eggs were acquired from a Bt-susceptible and Bt-resistant colony. The susceptible insects were from a laboratory strain reared with no Bt exposure for 18 years acquired from Benzon Research, Inc. (Carlisle, PA, USA). The resistant colony was collected from Wake Forest, North Carolina, USA, in non-Bt corn. Both colonies were reared on an artificial diet in the lab for 2 generations [37]. To minimize strain differences as much as possible, both the resistant and susceptible bollworms were reared using the same rearing methods in the same laboratory at NCSU on the same artificial diet under the same environmental conditions. Rearing conditions in the growth chamber were as follows: 14:10 L:D, 27 °C:24 °C L:D, and 60% RH, and the moths were mated to conspecifics for each colony. The artificial food was modified *H. zea* diet; the modification is described in Reisig et al. [37]. The same rearing methods were used as described in Reisig et al. for both resistant and susceptible colonies [37]. Cry1Ac susceptibility bioassays were also performed. The observed difference in susceptibility was 100-fold (the Cry1Ac LC_50_ was 43.79 µg/cm^2^ for the Bt-resistant strain and 0.43 µg/cm^2^ for the Bt-susceptible strain). All rearing and bioassay methods are described in detail in Lawrie et al. and Reisig et al. [22,37].

### 2.2. RNA Extraction

From the colonies, 5 Bt-resistant samples and 5 Bt-susceptible samples were prepared, each replicate sample consisting of 10 neonate *H. zea*. All neonates were unfed and lab-reared, preceding RNA extraction as described earlier. Neonates were mechanically homogenized into one DNAse and RNAse free tube for each sample within 6 h of emergence. From each pooled sample, total RNA was extracted using the RNeasy Mini Kit following the manufacturer’s protocol (Qiagen, Valencia, CA, USA). The purity of total RNA in each sample was then evaluated using an Agilent 2100 Bioanalyzer (Agilent Technologies, Santa Clara, CA, USA) by the NC State University Genomics Core Facility (Raleigh, NC, USA). Sequencing was conducted on samples that had a RNA Integrity Number >9.0.

### 2.3. RNA Sequencing

The NCSU Genomics Core Facility performed RNA-seq for this experiment. cDNA libraries for each sample (using 500 μg of total RNA each) were prepared for RNA-seq using the TruSeq RNA Library Prep Kit v2 (Illumina, San Diego, CA, USA) following the manufacturer’s protocol. Transcriptome sequencing was conducted on the NextSeq 500 System (Illumina, San Diego, CA, USA) using a paired end setting with a read length of 2 × 150 base pairs. A sequencing depth of >25 million reads per library was obtained using a High Output Flow Cell. A total of 10 mRNA libraries were then prepared, 5 each for resistant and susceptible. The SRA Toolkit v2.9.2 was used to convert raw reads to fastq files [38]. Fastq file read quality was then evaluated using the FastQC tool v0.11.7 [39]. A Phred score of >30 was required for the majority of the sequencing reads to establish a baseline for quality. Fastq files with suitable quality were then used for assembly and quality control steps.

### 2.4. Transcript Assembly and Differential Expression Analysis

Transcript assembly and quality control were performed by the NC State Bioinformatics Core (Raleigh, NC, USA). Reads were assembled with the StringTie program (v1.3.5, John Hopkins University, Baltimore, MD, USA) with 45,224 primary transcripts assembled into transcript set 1 using the *H. zea* reference genome [40]. The program Trinity (v2.8.4, Broad Institute and Hebrew University of Jerusalem, Jerusalem, Israel) was used to assemble an alternate set of transcripts (set 2) that were not aligned with StringTie in order to maximize transcript assemblies [41]. For transcript assembly, there were 149,108 transcripts assembled and processed using the Blobology program (v2.15.2, University of Edinburgh, Edinburgh, UK) to ascertain the presence of contaminants [42]. Transcripts matching to Lepidoptera were then saved (108,867 transcripts). From these, all ribosomal RNA transcripts were deleted. The remaining 108,841 transcripts were clustered with the Evigene program (v1.0, University of Indiana, IN, USA), which resulted in 34,059 transcripts in set 1 [43]. Transcript sets 1 and 2 were then combined and clustered using Evigene, resulting in 26,800 primary and 12,095 alternate transcripts. Primary and alternate sets were then analyzed with Blobology to check for contaminates again. Ribosomal RNA transcripts were also removed from these sets. Primary and alternate transcripts were subsequently clustered and combined with Evigene.

The TrimmoMatic sequence trimmer (v0.39, Max Planck Institute, Munich, Germany) was used to trim fastq files for each replicate for adapter sequence and quality [44]. The *H. zea* reference genome (NCBI) was used to map each trimmed file to the reference genome using HiSat2 [45]. StringTie was then used to assemble resulting mapped files to assemble RNA-seq alignments into potential transcripts. All transcript annotations from each replicate were then combined into one “expressed transcriptome” file. This was used to guide gene boundaries when calculating differential expression values (log2 fold change) between the susceptible and resistant strains with CuffDiff (v7.0, Cambridge, MA, USA) [46]. Statistical significance was determined using the Tuxedo Pipeline (in CuffDiff, which assigned transcript q-values, α = 0.05). Only statistically significant transcripts were included in later data analysis [47]. These results were then imported into the R statistical software platform for quality control checks and visualization of results [48]. The sequence of transcripts that were determined to be differentially expressed were extracted from the reference genome and used in BLASTn searches against insects to provide initial annotations. Quality control steps and data analysis were conducted with volcano plots, FPKM, boxplots, PCA plots, MDS plots, normalization, and heatmaps. These steps were performed to ensure replicates were of sufficient quality, mapping rate, and variation among replicates. Each replicate passed all quality control steps. After assembly and quality control were conducted for all transcripts, 6098 transcripts were identified as differentially expressed in this experiment. Of these, 3042 transcripts had higher expression in the susceptible strain, with 267 being found only in this strain. The remaining 3056 had higher expression in the resistant strain, with 323 being only expressed in the resistant strain. Blast2GO (v5.2.4) was utilized to annotate open reading frame assignments [49] and function. Gene ID and function were determined using BLASTx (E-value cut off 10^−5^), using lepidopteran taxonomy to filter results, using the nr and swissprot databases [49].

### 2.5. Data Analysis and Figure Construction

In order to categorize transcripts as “long non-coding RNAs”, all transcripts that were annotated as non-coding RNAs were separated from protein coding genes. GenBank was used to examine sequence length from these transcripts defined as non-protein coding genes by NCBI BLAST (BLASTx). All transcripts that were 200 base-pairs or greater were categorized as long non-coding RNAs (lncRNAs). A lncRNA is a non-protein-coding RNA over 200 base pairs in length [27]. There were 2 non-protein-coding RNA transcripts (Hzea.11974 and Hzea.13128) with <200 base pairs that were excluded. Appendix A depicts a visualization of the above process. Figures and tables for this paper were prepared using Microsoft Excel, PowerPoint, Word (2018), and SigmaPlot (v14.0, SigmaPlot, Systat Software, San Jose, CA, USA). All sequence alignments were conducted using mega BLAST (BLASTn), which also provided percent alignments, E-values, and query coverages [50]. The R statistical software platform was used to construct a volcano plot [48]. For each table, the fasta sequence for each of the lncRNAs identified was run through BLASTn to analyze candidate lncRNA validity. This was also done to determine if any candidate lncRNAs had been characterized since RNA-sequencing was conducted.

A pseudogene is an imperfect copy of a functional coding gene [51]. Often pseudogenes act as regulators for coding genes where they share ancestral traits (ex. cytochrome P450s and cytochrome P450 pseudogenes); we found evidence of this before for primary human liver cells held in short term culture and treated with the pesticides, DEET, and fipronil [51]. The pseudogene analysis was performed by selecting 5 lncRNAs with the highest log2 fold increase, 5 with the highest log2 fold decrease, and 2 at random from the only in resistant and 2 at random from the only in susceptible categories. Ten coding genes with known associations to Bt-resistance were selected (trypsin, serine protease, tetraspanin, cadherin, and beta-secretase). The coding gene with the highest log2 fold increase and highest log2 fold decrease for each of the coding gene types listed above was selected for comparison to the lncRNAs. NCBI BLASTn was used to align each lncRNA sequence to each coding gene sequence. E-value, percent identity, query coverage, and score values were assessed.

Genomic proximity also can indicate that two genes have a functional relationship with each other. Significant genomic proximity is defined as a distance *cis* or *trans* within 1000 kb [52]. In our previous work with primary short-term cultures of human liver cells, a number of differentially-regulated lncRNAs affected by exposure to the pesticides, DEET, and Fipronil, were found within significant genomic proximity to differentially regulated xenobiotic-metabolizing genes [51]. For proximity analysis, genomic scaffolds were assembled for analysis using the Integrative Genomics Viewer (IGV) program (v2.9.2, Broad Institute, Jerusalem, Israel) (Available online: http://software.broadinstitute.org/software/igv/home (accessed on 5 March 2021) to compare genomic proximities of lncRNAs to coding genes. First, the *H. zea* reference genome (NCBI) was loaded into IGV followed by all the differentially expressed transcripts (GTF file) from our RNAseq work. The same putative lncRNAs selected for the pseudogene analysis were used here for the genomic proximity analysis. Each of the lncRNAs was then located in IGV using scaffold numbers and coordinates. Protein coding genes within 1 million bp of each lncRNA were then located. The coding genes were identified for annotation using NCBI BLASTx and the lncRNAs examined as potential pseudogenes proximal to coding genes on each scaffold being studied. All proximal (within 1 million bp) coding genes and lncRNAs were aligned with NCBI BLASTn and analyzed.

## 3. Results

### 3.1. Characterization of Putative lncRNAs in H. zea

The workflow for the identification of possible lncRNAs is shown in Appendix A. A lncRNA was defined as an RNA sequence that annotated as a non-coding sequence using BLASTx and was over 200 base-pairs in length from RNAseq data obtained from replicated Bt-resistant and Bt-susceptible unfed neonates of *H. zea*. Only the statistically significant, differentially expressed lncRNAs between the resistant versus susceptible strains are reported, and these are in two categories: (i) those found in both strains and (ii) those found only in one strain. In the former, we were able to calculate a log2 fold increase (upregulation) or decrease (downregulation) in expression for the resistant strain compared to the susceptible strain (Appendix A). For lncRNAs only in the resistant or susceptible strains, a log2 fold difference between strains could not be calculated since the expression level was zero in the comparative strain (Appendix A).

There were 98 non-coding transcripts that were differentially expressed, of which 96 were ≥200 bp and classified as a lncRNA. Appendix A shows the range of sequence lengths for the differentially expressed lncRNAs. The lengths ranged from 297 to 6719 bp, with most sequences between 200–1000 bp. Additionally, lncRNAs that were upregulated in Bt-resistant bollworms were among the longest lncRNAs in this study, where eight were above 2500 bp in length; however, most of these lncRNAs did not have a large degree of log2 fold increase (all above 2500 bp had below a 2.0 log2 fold increase) (Appendix A). For downregulated lncRNAs, there was only one lncRNA above 2500 bp in length, where again the log2 fold decrease was below 2.0 (Appendix A). For the only in resistant and only in susceptible groups, there was one lncRNA for each category over 2500 bp in length; however, the other lncRNAs in these categories were all below 1000 bp (Appendix A).

### 3.2. Differential Expression of Putative Long Non-Coding RNAs in Bt-Resistant H. zea

Log2 fold changes for putative lncRNAs with increased expression in resistant insects ranged from 4.88 to 0.38, and decreased expression ranged from 4.41 to 0.44 (Figure 1). Those lncRNAs with an increased expression not only had a higher maximum log2 fold change in comparison to those with decreased expression (4.88 log2 fold increase vs. 4.44 log2 fold decrease, respectively) but also had a consistently higher number of lncRNAs with higher degrees of expression. There were 33 upregulated lncRNAs above 1.0 log2 fold change and 13 downregulated lncRNAs above 1.0 log2 fold change.

In order to further compare the differences in lncRNA expression between the two strains of *H. zea*, different thresholds of expression were compared (Figure 1; Appendix A). When examining the 96 differentially expressed lncRNAs identified, there were 58 with increased expression in the Bt-resistant strain, 24 with decreased expression, 10 only in the resistant strain, and four only in the susceptible strain (Figure 1, Appendix A). Using a threshold of ≥1.0 log2 fold change, there were 33 upregulated lncRNAs and 13 downregulated lncRNAs (Figure 1). Using a threshold of ≥2.0 log2 fold change, there were 17 upregulated lncRNAs and six downregulated lncRNAs (Figure 1). The five lncRNAs with the highest log2 fold change lncRNAs in either direction are shown in Figure 1 and Appendix A. The top five upregulated transcripts were LOC113506107 with a 4.88 log2 fold increase, LOC113508874 with a 4.65 log2 fold increase, LOC110372550 at 3.9, LOC110380503 at 3.5, and LOC110371745 at 3.44. The top five downregulated transcripts were LOC110373805 at a 4.41 log2 fold decrease, LOC110373534 at 3.68, LOC110382662 at 3.46, LOC110383440 at 3.05, and LOC110369725 at 2.83. Overall, not only did the Bt-resistant strain have a higher number of upregulated lncRNAs but also the magnitude of log2 fold changes were consistently higher.

### 3.3. Functional Characterization of Long Non-Coding RNAs in Bt-Resistant H. zea

In order to determine whether there were any pseudogenes present in our data, NCBI BLASTn was used to compare differentially expressed lncRNA sequences to differentially expressed protein-coding genes with functions known to be important in Bt-resistance. Five lncRNAs with the highest log2 fold increase, five with the highest log2 fold decrease, two found only in the resistant strain, and two only in the susceptible strain were compared by NCBI BLASTn with five coding genes with the highest log2 fold increase and five with the highest log2 fold decrease for a serine protease, ABC transporter, trypsin, secretase, and tetraspanin. These proteins have functions known to be important in Bt-resistance (Figure 2, Appendix A). A majority of the sequences did not have any significant alignments. All results are depicted in the Appendix A. The best pseudogene candidate was lncRNA LOC110369725 and cadherin XJ-r15 (Figure 2). The BLASTn alignment was as follows: E-value = 0, percent identity = 99.07%, query coverage = 81%, max score = 950, total score = 1002. A BLASTx alignment of XJ-r15 showed multiple exons and introns. The section that was translated into cadherin did not align with LOC110369725. The putative lncRNA aligned elsewhere on the XJ-r15 cadherin gene sequence.

To examine proximity relationships that might be important in Bt-resistance, we identified the genome scaffolds that contained the five lncRNAs with the highest log2 fold increase, five with the highest log2 fold decrease, two found only in the resistant, and two only in the susceptible bollworm strains (Figure 3). We then located all coding genes within significant proximity upstream and downstream of each lncRNA, and these were annotated by NCBI BLASTx. Even though proximity is defined as 1 million base pairs *cis* and *trans* from the lncRNA, proximity measurements were smaller because of the smaller scaffold size. The results of this analysis are shown in Figure 4A–E and Appendix A. A wide variety of coding genes were found in genomic proximity to the lncRNAs we examined. Most interesting, known Bt-resistance associated genes were found in genomic proximity to a number of these lncRNAs. These were a CYP (Hzea.12028, CYP6B7, CYP6B6, CYP6B2) (Figure 4A), an ABC transporter (Hzea.20383, ABCC3, ABCC2, ABCC1) (Figure 4B), and a serine protease (Hzea.7824, LOC110382673, serine protease snake-like) (Figure 4C). Among the lncRNAs we examined, there were also lncRNAs that did not have any genomic proximities (Figure 4D) and those that were uncharacterized or unrelated to Bt-resistance coding genes (Figure 4E). Each proximal Bt-resistance associated gene was less than 100 kb up- or downstream from the lncRNA (Figure 4A–C). The proximal CYP was 7.868 kb from lncRNA LOC11350610 (this lncRNA was upregulated) (Figure 4A), the proximal ABC transporter 50.672 kb from lncRNA LOC110369725 (this lncRNA was downregulated) (Figure 4B), and the serine protease 0.646 kb from lncRNA LOC110382674 (this lncRNA was found only in the resistant strain) (Figure 4C). The lncRNA presented in Figure 4D was downregulated, and the lncRNA presented in Figure 4E was found only in the susceptible strain.

We also looked for putative pseudogenes among the proximal coding genes and lncRNAs (Figure 3) with NCBI BLASTn comparisons. There were no significant alignments found to Bt-resistance associated genes (Figure 4A–E). However, there was a lncRNA (LOC110372708) that aligned to a previously characterized prostaglandin pseudogene (Appendix A) using the same methodology used to find the putative cadherin pseudogene (Figure 2). The lncRNAs that had significant proximities to Bt-resistance associated genes (Figure 4A–C) were also aligned with each other using BLASTn to see if they had any significant similar regulatory potential. Each of these three lncRNAs did not show any significant alignment. All of the scaffolds studied were less than 1 million bp in length on either side of the lncRNA. Therefore, it is possible that other significant proximities exist that could not be detected in our research.

## 4. Discussion

### 4.1. Characterization of Long Non-Coding RNAs in H. zea

We successfully identified 96 differentially expressed non-protein-coding sequences in *H. zea* that are candidate lncRNAs. This study focused only on statistically significantly differentially expressed lncRNAs between a Bt-resistant and Bt-susceptible strain of *H. zea* (Figure 1, Appendix A). These 96 were those that were statistically significantly differentially expressed (Tuxedo suite pipeline determined significance) [22]. In the diamondback moth, *Plutella xylostella*, 3324 lncRNAs were identified by RNA-seq [53]. In the human genome, 100,000 lncRNAs have been found where differential expressions of several lncRNAs were linked to a wide variety of pathologies [54]. Therefore, it is likely that there are additional lncRNAs present in *H. zea*; they simply were not significantly differentially expressed between the Bt-resistant and Bt-susceptible strains and were not considered in our study. It is possible that the putative lncRNAs that were differentially expressed in this study are important in the mechanism of Bt-resistance, and/or differential expression could be caused by insect strain differences. The Bt-resistant strain was obtained from the field, and the Bt-susceptible strain was reared in the laboratory for many generations on an artificial diet. To minimize strain differences, both the resistant and susceptible bollworms were reared in the same lab under the same conditions with the same artificial diet for two generations, and the study was conducted on unfed neonates. This approach minimizes any possible differences in developmental rates after hatching and diet preferences. Furthermore, we were measuring constitutive gene expression levels essentially at the time of hatching and before feeding effects can impact development. It should be noted that Cry2Ab resistance in *H. zea* is present in North Carolina, and the resistant colony used in this study was Cry1Ac and Cry2Ab resistant; however, only Cry1Ac susceptibility bioassays were performed for this work [11]. This bioinformatics approach was a first step in identifying and characterizing the non-coding *H. zea* genome. More research will be needed to understand fully the function of the non-coding RNAs found in this study.

When we examined lncRNA base-pair length (Appendix A), the majority of transcripts ranged from 200 to 1000 bp, but there were several lncRNAs above 4000 bp. LncRNAs have the capability to exhibit secondary structures, such as a stem loop and cloverleaf structure [55]. The type of secondary or tertiary lncRNA structure can provide evidence of its mechanism for regulating a protein-coding gene. In the case of the lncRNA HOTAIR (which is 2.2 kb), it was discovered that double stem loops at the 5′ and 3′ ends of the lncRNA are important in chromatin remodeling, a regulatory function of HOTAIR [55]. Another lncRNA, MALAT1 (over 8.7 kb), was discovered to exhibit a cloverleaf secondary structure at the 3′ end, similar to cloverleaf functions in tRNAs. These are linked to subcellular localization [55]. Sequence length is useful to consider because it is a predictor of possible lncRNA secondary structure. In this study, there were nine lncRNA candidates above 3 kb (with log2 fold changes below 2.0 in either direction). This could indicate that these particular lncRNAs have secondary structures important in the mechanism of regulating protein-coding genes. Further modeling is needed of these >3 kb, differentially expressed lncRNAs in Bt-resistant bollworms to determine their 3-D structure and possible mechanism of action.

### 4.2. Differential Expression of Long Non-Coding RNAs in Bt-Resistant H. zea

In the Bt-resistant strain, there were a significantly greater overall number of lncRNAs with increased expression levels (58 up- vs. 24 down-regulated) (Figure 1). Additionally, when examining higher magnitudes of log2 fold change, there were both greater numbers of lncRNAs with high degrees of log2 fold change and an overall average greater magnitude of log2 fold change (Figure 1). It is possible that the magnitude of log2 fold change does not signify importance to Bt-resistance. However, the high degrees of lncRNA upregulation in Bt-resistant *H. zea* is one argument for their functional role in Bt-resistance. For example, a specific lncRNA upregulated in this experiment may be acting as an enhancer for a coding gene involved in Bt-resistance, such as a serine protease or an ABC transporter. One major functional role of lncRNAs is enhancing coding gene expression [27,51]. In the pink bollworm, *Pectinophora gossypiella*, it was discovered that the lncRNA pgCad1 lncRNA is a specific enhancer of the cadherin gene pgCad1 [36]. When pgCad1 lncRNA was silenced using siRNA, the expression levels of pgCad1 (and also Bt-susceptibility) were significantly reduced [36]. Therefore, it is possible that an upregulated lncRNA identified in this study is linked to upregulation of a Bt-resistance coding gene. LncRNAs can also act as repressors for coding genes, causing downregulation [27].

It is important to note that a lab-reared Bt-susceptible strain was used as a reference strain in these RNA-seq experiments. Ideally, a field-collected Bt-susceptible strain of *H. zea* would be used as a reference for a field-collected Bt-resistant strain. However, due to the prevalence of Bt-resistance in wild *H. zea* populations, completely susceptible insects are difficult to collect in sufficient numbers to establish a colony easily. In order to maximize the comparison of the Bt-resistant insects with the susceptible strain, we used the lab-reared susceptible strain (Benzon), which has been used before by multiple research groups [22,56,57]. The advantage of this experimental design, we were able to compare gene expression to the Benzon strain (commercially available as a reference strain) with minimal rearing in the lab of the resistant field strain. The more generations of rearing of the resistant insects in the lab, the greater the chance they are different from the insects from the field. Research is needed to develop a better comparative, Bt-susceptible field strain of the bollworm.

### 4.3. Predicting Function of Long Non-Coding RNAs in Bt-Resistant H. zea

Pseudogenes are imperfect copies of parent coding genes; often called “genomic fossils”. They can play a key role in the regulation of their parent genes (ex., a lncRNA derived from a CYP pseudogene regulates a CYP mRNA) [51,58]. Certain pseudogenes are known to transcribe for non-coding RNAs, such as lncRNAs [59,60]. The non-coding RNAs derived from pseudogene sequences can then act as specific regulators of parent coding genes where the derived non-coding RNAs tend to have high sequence similarity to parent genes [59,60]. Previously thought to be untranslated, some pseudogenes are translated in humans; it is unknown if this occurs in insects [59,60]. Pseudogenes can be located anywhere within a genome but often are adjacent to their functional parent genes [51,58]. In this study, we were able to identify one candidate for a pseudogene that may be regulating an important Bt-resistance associated gene (putative lncRNA LOC110369725 and cadherin XJ-r15) where there was a high degree of sequence similarity (Figure 2). It is possible that since no BLASTx alignment was found that putative lncRNA LOC110369725 is derived from a pseudogene that is not translated into protein and only interacts with XJ-r15 cadherin at the gene level. This potential functional categorization was performed using BLAST only. More work is needed to support this hypothesis. One regulatory function of pseudogenes is their processing into piRNAs (piwi-interacting RNAs), where the pseudogene, once spliced into smaller piRNAs, functions in RNAi-mediated gene silencing [61]. It could be that the proposed pseudogene LOC110369725 is being processed into piRNAs prior to regulatory interactions with cadherin XJ-r15. More research is needed to confirm this hypothesis. The characterization of pseudogene function is a rapidly evolving field where new data are changing our understanding of pseudogenes as new research is conducted.

There are likely many more pseudogenes present within *H. zea*. The analysis performed in this study focused on those potential pseudogenes that were differentially expressed in Bt-resistant insects. Pseudogenes unrelated to Bt-resistance are likely to be present in the genome. For example, another pseudogene annotated as a prostaglandin reductase pseudogene was discovered proximal to one of the differentially expressed lncRNAs we examined (Appendix A). Further identification of pseudogenes in *H. zea* would provide greater insight into the genomic functioning of this important pest species and the potential use of pseudogenes as targets for gene editing and pest management. Characterization of pseudogenes in insects would also be useful in understanding the evolution of genes throughout an insect’s natural history.

Significant genomic proximity (within 1,000,000 bp) can be indicative of a relationship between two sequences [51,52,62]. In this study, we examined the genomic scaffolds of some of the greatest differentially expressed lncRNAs. Several significant proximities were discovered for a wide variety of coding genes (Figure 4A–E, see also Appendix A). Interestingly, three putative lncRNAs with significant proximities to coding genes related to resistance, i.e., CYP, an ABC transporter, and a serine protease (Figure 4A–C). Changes in the expression of CYPs have been linked to pyrethroid resistance in *H. zea*; it is possible that the differential expression of CYPs in this data set is related to pyrethroid resistance [63]. The putative lncRNAs presented in this study could be linked to the regulation of CYP genes that are involved in pyrethroid resistance. While pyrethroid resistance in *H. zea* has been documented in the southeastern USA, pyrethroid resistance was not assayed for in this study; it is unknown if the Bt-resistant strain was also pyrethroid-resistant [64]. In particular, the serine protease gene was within 1000 bp of the proximal lncRNA (Figure 4C). It is possible that due to the significant proximities to these coding genes, the lncRNA LOC113506107 (Figure 4A), lncRNA LOC110369725 (Figure 4B), and lncRNA LOC110382674 (Figure 4C) act as regulators in some capacity to the proximal coding genes with functions at least associated with Bt-resistance. However, it is unlikely that these particular lncRNAs are pseudogenes due to no significant alignments being present after BLAST. In the pink bollworm, *P. gossypiella*, a specific lncRNA that is intronic to a cadherin gene, has been established as an enhancer of that cadherin [36]. Additionally, in human liver cell models, we found a link (high level of similarity) between a wide variety of lncRNAs and proximal coding genes important in drug metabolism [51]. We did not find this to be the case in our bollworm study. By identifying specific regulators of coding genes important to Bt-resistance, it is possible that novel means of resistance management may be developed. For example, RNAi-mediated silencing of a lncRNA could be used to enhance the expression of a cadherin (or another type of Bt-receptor), increasing Bt-susceptibility. Bt-susceptibility has successfully been altered before using this technique targeting a lncRNA regulating the cadherin gene in *P. gossypiella* [36]. Additionally, gene-editing approaches that target non-coding genes might be more useful in insect resistance management and insect control than targeting coding genes (with greater non-target effects, such as RNAi impacting the target species and other closely related species). In plants, for example, lncRNAs have a high degree of intraspecies conservation with greater sequence diversity between species [65]. Targeting lncRNAs for resistance management may be more species-specific than targeting coding genes; however, much more research and characterization of lncRNAs in insects is needed. This study has only identified a small number of lncRNAs that could be important to Bt-resistance but is a step towards a greater understanding of how lncRNAs work in insects in general and in Bt-resistance.

## 5. Conclusions

This study examined the differential regulation of putative lncRNAs in a field Bt-resistant strain of unfed neonates of the bollworm, *H. zea*. Overexpression of lncRNAs in other lepidopteran models has been correlated to chemical and Bt-insecticide resistance [31,36]. This study provides a comprehensive list of lncRNAs in *H. zea* associated with Bt-resistance and predicts potential regulatory roles therein for the first time. We characterized a possible pseudogene and multiple examples of genomic proximity between differentially regulated lncRNAs and differentially regulated protein-coding genes where the protein function is a known mechanism for Bt-resistance. It is likely that additional lncRNAs to those that were differentially expressed between the Bt-resistant and Bt-susceptible strains are present.

## Figures and Tables

**Figure 1 insects-13-00012-f001:**
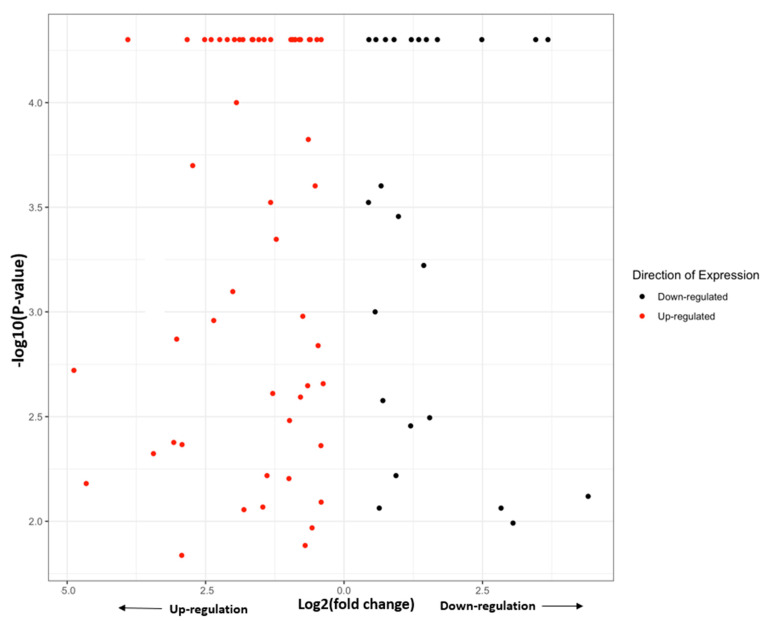
Log2 fold change in differentially expressed up-(red dots) or down-(black dots) regulated putative lncRNAs. The X-axis depicts the magnitude of log2 fold change in Bt-resistant *H. zea* where values on the left of 0.0 indicate the fold upregulation and on the right of 0.0 the log2 fold downregulation in the Bt-resistant strain of *H. zea*. The Y-axis is the *p*-values for statistical significance.

**Figure 2 insects-13-00012-f002:**
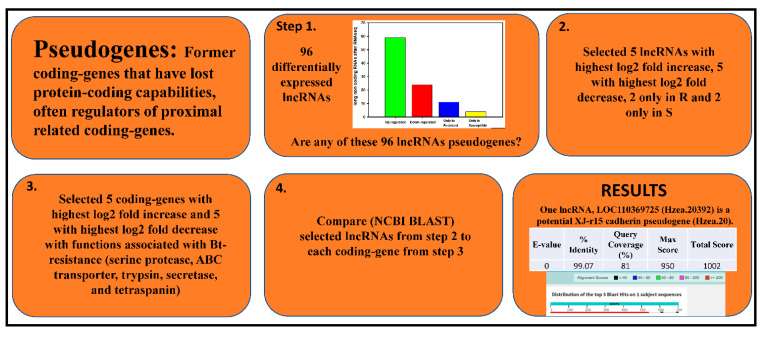
Workflow to identify statistically differentiated lncRNAs as putative pseudogenes.

**Figure 3 insects-13-00012-f003:**
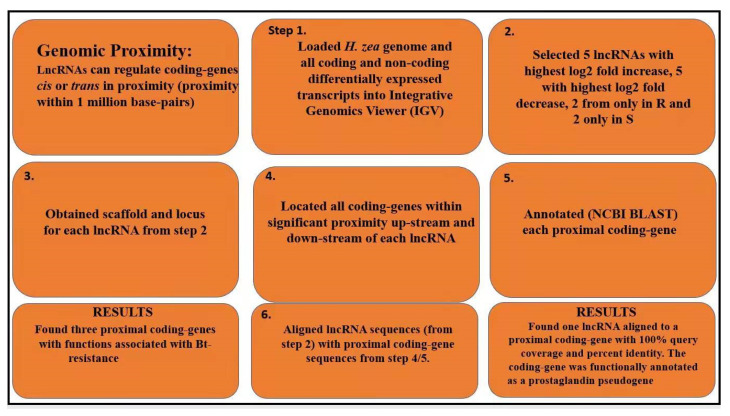
Workflow for identifying statistically differentiated lncRNAs coding genes *in toto* and those with functions known to have a role in Bt-resistance that are proximal to statistically differentiated lncRNAs. Proximity measurements were limited by the size of the scaffolds, even though proximity is defined as 1 million base pairs *cis* and *trans* from the lncRNA. For each proximal coding gene and lncRNA, a BLASTn alignment was also conducted to assess potential pseudogenes.

**Figure 4 insects-13-00012-f004:**
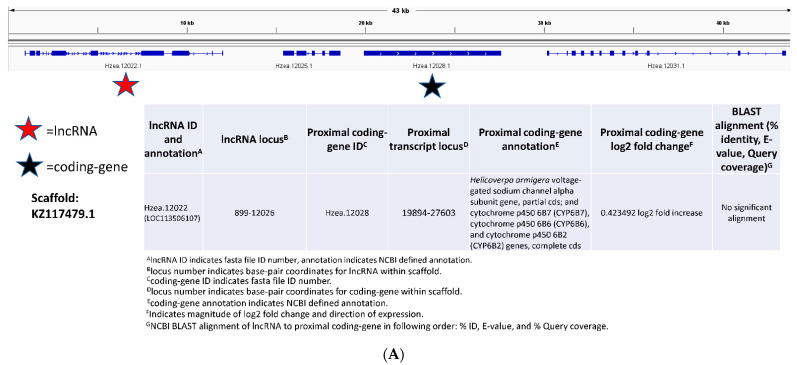
Genomic scaffold for lncRNAs and identification of proximal protein-coding genes. The scaffold at the top of each (**A**–**E**) depicts the range of the scaffold in kilobases (kb). The bars in blue indicate sequences present on each scaffold, with gene ID numbers below each. The red stars indicate a lncRNA; the black stars indicate a protein-coding gene. The scaffold ID number is placed directly below the legend on the left side. The table below the scaffold includes the following information about the lncRNA and coding genes found in the scaffold from left to right: the lncRNA ID number and annotation, lncRNA loci coordinates, gene ID number of the proximal coding gene, coding gene loci coordinates, coding gene annotation (NCBI defined), coding gene log2 fold change, and BLASTn alignment results (% identity, E-value, and query coverage) comparing the lncRNA and the protein-coding gene. Each subfigure depicts the following: (**A**), lncRNA LOC113506107 proximal to a CYP coding gene; (**B**), lncRNA 110369725 proximal to an ABC transporter coding gene; (**C**), lncRNA LOC110382674 proximal to a serine protease coding gene; (**D**), lncRNA LOC110373534 with no significant proximal coding genes; and (**E**), lncRNA LOC110383387 proximal to non-Bt-associated coding genes. All other proximity analyses can be found in Appendix A.

## Data Availability

All sequencing data have been uploaded to NCBI; the BioProject ID number is PRJNA761717. Please contact authors if further questions arise at any time.

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
