# Peer review of "Characterization of Long Non-Coding RNAs in the Bollworm, Helicoverpa zea, and Their Possible Role in Cry1Ac-Resistance"

_insects, 2021, doi:10.3390/insects13010012_

Round 1

Reviewer 1 Report

The manuscript “Characterization of long non-coding RNA in the bollworm, Helicoverpa zea, and their possible role in Cry1Ac-resistance” was reviewed. The authors analyzed some putative lncRNAs from Cry1Ac-resistant Helicoverpa zea strain through RNA-seq and also found some lncRNAs possible involved in the pest resistance. This study is very interesting and it is the first step for identifying the non-coding H. zea genome, which would promote the function of the non-coding RNAs in H. zea in the future. However, some minor points need to be addressed before it is considered to be accepted to publish,

  1. In abstract, the authors state “More research is needed to study Ag pest lncRNAs and their role in insect pest function and resistance”. What is Ag pest….? I suggest delete this sentence due to the nonsense.
  2. In introduction, “Research is minimal by comparison in insects, even less for agricultural pests, and there has been almost no work on their role in pesticide resistance.” Actually, there is a recent study on the agricultural insect, Plutella xylostella, demonstrating an important role of the lncRNAs, so few studies, instead of “no work”, were demonstrated. The related reference could be cited.
  3. Fig. 1 in the results could be removed to the supplementary materials and the statements are combined to the appropriate place in M&M.
  4. What’s the origin of the H. zea Bt-resistant strain? This should be added to the 4.1 section.

Reviewer 2 Report

The authors characterized a possible pseudogene and multiple examples of genomic proximity between differentially regulated lncRNAs in Helicoverpa zea, evaluating their possible role in Cry1Ac-resistance.

The work is suitable for publication after an accurate revision:

     - Introduction: You should introduce here the concepts of pseudogenes and genomic proximity in the context of lncRNAs, so that you can avoid using citations in the results section.

  • Simple summary: I think there’s a misunderstanding of what is the function of a simple summary, which is “to describe the work simply and concisely to the public, without using technical terms”. Please don’t use technical terms (e.g. “statistically significant”) and abbreviations. I paste here the description found on Insects (MDPI) site, which should guide you to rewrite that paragraph.

It is vitally important that scientists are able to describe their work simply and concisely to the public, especially in an open-access on-line journal. The simple summary consists of no more than 200 words in one paragraph and contains a clear statement of the problem addressed, the aims and objectives, pertinent results, conclusions from the study and how they will be valuable to society. This should be written for a lay audience, i.e., no technical terms without explanations. No references are cited and no abbreviations. Submissions without a simple summary will be returned directly. Example could be found at https://www.mdpi.com/2075-4450/11/8/508.

  • Transcripts assembly and quality control: This section should be called “Transcripts assembly and differential expression analysis”. Please specify what is the purpose of the de novo transcriptome assembly described in the first part. Why do you assembled the transcriptome if later that transcriptome is not used for the differential expression analysis (which is based on reads aligned on the genome)? Moreover, I guess that trimmomatic is also used before you performed the de novo transcriptome assembly.   
  • Data analysis and figure construction: You don’t need to repeat : “E-value, percent identity, query coverage, and score values were assessed.” Every time. You can just write it once, when you cite the BLAST paper. Also “We selected 5 lncRNAs with the highest log2 fold increase, 5 with the highest log2 fold decrease, and 2 at random from the only in resistant and 2 at random from the only in susceptible categories.” Is repeated at the end of the section.
  • All figures: Figures are of low quality. Please use high quality (e.g. 300 DPI) figures.
  • Figure 2: You can also use a single volcano plot which is more informative than all these histograms.
  • Figure 6: I suggest you to move figure 6 to supplementary material.
